# Evaluation of the Intracellular Signaling Activities of κ-Opioid Receptor Agonists, Nalfurafine Analogs; Focusing on the Selectivity of G-Protein- and β-Arrestin-Mediated Pathways

**DOI:** 10.3390/molecules27207065

**Published:** 2022-10-19

**Authors:** Masahiro Yamaguchi, Kanako Miyano, Shigeto Hirayama, Yusuke Karasawa, Kaori Ohshima, Eiko Uezono, Akane Komatsu, Miki Nonaka, Hideaki Fujii, Keisuke Yamaguchi, Masako Iseki, Masakazu Hayashida, Yasuhito Uezono

**Affiliations:** 1Department of Pain Medicine, Juntendo University Graduate School of Medicine, 2-1-1 Hongo, Bunkyo-ku, Tokyo 113-8421, Japan; 2Department of Pain Control Research, The Jikei University School of Medicine, 3-25-8 Nishi-Shimbashi, Minato-ku, Tokyo 105-8461, Japan; 3Medical Affairs, Pfizer Japan Inc., 3-22-7 Yoyogi, Shibuya-ku, Tokyo 151-0053, Japan; 4Division of Cancer Pathophysiology, National Cancer Center Research Institute, 5-1-1 Tsukiji, Chuo-ku, Tokyo 104-0045, Japan; 5Laboratory of Medicinal Chemistry and Medicinal Research Laboratories, School of Pharmacy, Kitasato University, 5-9-1 Shirokane, Minato-ku, Tokyo 108-8641, Japan; 6Medical Affairs, Viatris Pharmaceuticals Japan Inc., 5-11-2 Toranomon, Minato-ku, Tokyo 105-0001, Japan; 7Department of Anesthesiology and Pain Medicine, Faculty of Medicine, Juntendo University, 2-1-1 Hongo, Bunkyo-ku, Tokyo 113-8421, Japan; 8Supportive and Palliative Care Research Support Office, National Cancer Center Hospital East, 6-5-1 Kashiwanoha, Kashiwa-shi, Chiba 277-8577, Japan

**Keywords:** analgesic, κ-opioid receptor, G-protein-biased agonists, nalfurafine, bias factor

## Abstract

Opioid receptors (ORs) are classified into three types (μ, δ, and κ), and opioid analgesics are mainly mediated by μOR activation; however, their use is sometimes restricted by unfavorable effects. The selective κOR agonist nalfurafine was initially developed as an analgesic, but its indication was changed because of the narrow safety margin. The activation of ORs mainly induces two intracellular signaling pathways: a G-protein-mediated pathway and a β-arrestin-mediated pathway. Recently, the expectations for κOR analgesics that selectively activate these pathways have increased; however, the structural properties required for the selectivity of nalfurafine are still unknown. Therefore, we evaluated the partial structures of nalfurafine that are necessary for the selectivity of these two pathways. We assayed the properties of nalfurafine and six nalfurafine analogs (SYKs) using cells stably expressing κORs. The SYKs activated κORs in a concentration-dependent manner with higher EC_50_ values than nalfurafine. Upon bias factor assessment, only SYK-309 (possessing the 3*S*-hydroxy group) showed higher selectivity of G-protein-mediated signaling activities than nalfurafine, suggesting the direction of the 3*S*-hydroxy group may affect the β-arrestin-mediated pathway. In conclusion, nalfurafine analogs having a 3*S*-hydroxy group, such as SYK-309, could be considered G-protein-biased κOR agonists.

## 1. Introduction

Opioid analgesics such as morphine are widely used to improve various forms of pain, including chronic pain, perioperative pain, and cancer pain [1,2,3,4]. ORs belong to the G-protein-coupled receptor (GPCR) family [5] and are classified into three subtypes (μORs, δORs, and κORs), where each type of OR is associated with analgesic effects. Current opioid analgesics mainly bind to μOR to exert their analgesic effects [6]. The typical OR agonists, such as morphine, activate μORs and show strong antinociceptive effects. Therefore, μOR agonists have been used as effective analgesics to alleviate acute and severe chronic pain. However, their use is sometimes restricted by unfavorable side effects, such as opioid dependence, tolerance, constipation, itch, or respiratory depression [7,8,9], while increasing opioid misuse and opioid-associated mortality have also been observed [10]. In fact, approximately 20% of patients with chronic pain using opioids were abusing them [11]. Thus, the investigation of novel opioids as an adjuvant to μOR-targeting analgesics, which have sufficient analgesic effects and lower side effects, is necessary.

Besides selective μOR agonists, several selective δOR or κOR compounds have been investigated [12,13]. In particular, κORs are ubiquitously and widely expressed throughout the central nervous system and activated by opioid peptides, such as dynorphins [14,15,16,17]. The κOR/dynorphin system is linked to reducing pruritus as a potential therapeutic action [18]. In practice, it has been proven that κOR agonists were efficacious in treating intractable itch or pruritus [18,19,20,21]. Moreover, like other ORs, activating the κOR promotes antinociception, and κOR agonists produce an analgesic effect without respiratory depression that is often seen with μOR agonists [22]. Therefore, κOR agonists have been investigated as alternatives to μOR agonists for pain treatment [1]; however, their therapeutic potential is limited by negative side effects such as sedation, motor incoordination, dysphoria, and psychotomimesis [23,24,25,26].

After the ligand conjugates to ORs, including κORs, the intracellular signaling from the ORs is transmitted through two major pathways: one is a G-protein-mediated pathway induced by decreasing the intracellular cAMP levels and is required for analgesia, and the other is a β-arrestin-mediated pathway that affects the recruitment of β-arrestin and is associated with unfavorable side effects [27,28,29,30,31,32,33]. Moreover, there is evidence which calls into question the concept of developing G-protein-biased μOR agonists as a strategy for developing safer opioid analgesic drugs [34]. Thus, there is no definite conclusion that the G-protein-mediated pathway is “good”, and the β-arrestin-mediated pathway is “bad”, which is still under discussion. However, to develop safer and more effective opioid analgesics, investigating G-protein-biased agonists or those with a pharmacological profile that prioritize activation of the G-protein-mediated pathway over the β-arrestin-mediated pathway may be desirable [35,36]. From these viewpoints, some molecules were investigated and indicated as G-protein-biased agonists [37,38]. Among them, oliceridine was evaluated by intravenous administration in clinical studies and was approved as the first G-protein-biased μOR agonist that could be prescribed [39]. Subsequently, the expectations in researching the selective activation of G-protein- or β-arrestin-mediated pathways via κORs have also increased. In fact, the analgesic therapeutic outcomes of G-protein-mediated signaling through κORs have been investigated in some studies [40,41,42]. These results suggest that G-protein-mediated signaling after binding to the κOR could be involved in analgesic effects. In contrast, β-arrestin-mediated signaling could be related to unfavorable effects [40]. Thus, κOR agonists with a pharmacological profile of selectively activating the G-protein-mediated pathway over the β-arrestin-mediated pathway may be promising targets for more effective analgesics with fewer unfavorable effects [35,36,43].

Nalfurafine was discovered in 1998 by Dr. Hiroshi Nagase in Japan and was found to promote antinociception without aversion [44], acting as a selective κOR agonist [45]. Nalfurafine also reduced pruritus [46] and was launched as nalfurafine hydrochloride (Remitch^®^), an antipruritic drug in Japan [47,48]. Nalfurafine is the first and, currently, only available selective κOR agonist for the treatment of intractable pruritus suffered by patients on hemodialysis [49,50]. In addition, κOR agonists, including nalfurafine, may not be addictive because they do not induce euphoria, nor do they promote increases in dopamine release as abused drugs do [51,52,53]. In fact, nalfurafine, initially, was developed as an analgesic; however, its indication was changed from an analgesic to an antipruritic drug because the analgesic and sedative effects were not well separated [47,54]. Even now, no published clinical evidence indicates the efficacy of nalfurafine for the treatment of pain, and nalfurafine is no longer approved as an analgesic. Therefore, in our previous study, we investigated the affinity of nalfurafine and its analogs for κORs using binding assays and revealed the structure–activity relationship (SAR) of nalfurafine for κORs [55]. The differences in affinity of each nalfurafine analog for κORs were observed; however, how the structural properties of nalfurafine and its analogs affect the selectivity of the G-protein- and β-arrestin-mediated pathways remains unknown. Accordingly, to mitigate the negative effects of κOR agonists, especially nalfurafine analogs, and effectively utilize them for the treatment of pain, further investigation into the relationship between the structural features of nalfurafine and G-protein-/β-arrestin-mediated signaling activities is required.

Therefore, in the present study, we aimed to evaluate the partial structures of nalfurafine and six nalfurafine analogs (SYK-160, -186, -245, -308, -309, and -406; Figure 1) that are necessary for the selectivity of G-protein- and β-arrestin-mediated pathways. We used the CellKey^TM^, GloSensor^®^ cAMP, and PathHunter^®^ β-arrestin recruitment assays in cells stably expressing κORs. We estimated the G-protein-biased factor of each nalfurafine analog in comparison with nalfurafine to contribute to the development of more useful opioid analgesics.

## 2. Results

### 2.1. The Effects of Nalfurafine and Nalfurafine Analogs on the Functions of κORs Using the CellKey^TM^ System

We evaluated the effects of nalfurafine and six nalfurafine analogs (SYK-160, -186, -245, -308, -309, and -406) on κOR activities using the CellKey^TM^ system in HEK293 cells stably expressing Halotag^®^-κOR/pGS22F. The CellKey^TM^ system detects the activities of GPCRs, including κORs, as changes in cellular impedance [56]. The E_max_ and EC_50_ values were calculated, and we compared them between nalfurafine and its analogs. Nalfurafine and each nalfurafine analog activated κORs in a concentration-dependent manner; however, none of the six analogs exhibited E_max_ (%) values higher than those of nalfurafine (Figure 2). In contrast, the log EC_50_ (M) values of SYK-186 (removed the 3-hydroxy group from nalfurafine in Group A), -245 (removed the 3-hydroxy group and 4,5-ether bridge from nalfurafine, and converted the benzene ring to a cyclohexene ring in Group B), -308 (removed the 4,5-ether bridge from nalfurafine, converted the benzene ring to a cyclohexene ring, and added a 3*R*-hydroxy group in Group B), and -406 (removed the 3-hydroxy group and 4,5-ether bridge from nalfurafine in Group A) were significantly increased compared to those of nalfurafine (Table 1).

### 2.2. The Effects of Nalfurafine Analogs on the Intracellular cAMP Levels Evaluated Using the GloSensor^®^ cAMP Assay

We evaluated the actions of test compounds on κOR-induced G-protein signaling by measuring the intracellular cAMP levels using HEK293 cells stably expressing Halotag^®^-κOR/pGS22F. The E_max_ and EC_50_ values were calculated using nalfurafine as a positive control, and the six nalfurafine analogs caused a concentration-dependent decrease in cAMP levels (Figure 3). In detail, there were no nalfurafine analogs that showed E_max_ (%) values higher than nalfurafine, and the log EC_50_ (M) values of all nalfurafine analogs were significantly increased compared to those of nalfurafine (Table 1). These results suggested that the six nalfurafine analogs used in this study showed lower G-protein-mediated signaling activities than nalfurafine.

### 2.3. Effects of Nalfurafine Analogs on β-Arrestin Recruitment Using the PathHunter^®^ Recruitment Assay

To evaluate the actions of nalfurafine and six nalfurafine analogs on κOR-induced β-arrestin signaling, the PathHunter^®^ β-arrestin recruitment assay was performed using U2OS cells stably expressing κORs (DiscoverX, Fremont, CA, USA). Nalfurafine and each nalfurafine analog induced β-arrestin recruitment to κORs in a concentration-dependent manner (Figure 4). We calculated the E_max_ and EC_50_ values of these compounds, and no nalfurafine analogs showed E_max_ (%) values higher than nalfurafine. However, the log EC_50_ (M) values of all nalfurafine analogs were increased, except SYK-160, which has a non-significant increase compared to those of nalfurafine.

### 2.4. The Selectivity of G-Protein- and β-Arrestin-Mediated Pathways (G-Protein-Biased Factors)

The selectivity of G-protein- and β-arrestin-mediated pathways is indicated as a G-protein-biased factor representing the ratio of the value for G-protein signaling divided by that of β-arrestin signaling [57]. A biased factor > 1 indicates a preference for G-protein-mediated signaling activities, whereas a biased factor < 1 indicates a preference for the recruitment of β-arrestin as β-arrestin-mediated signaling activities compared to the control compound [56]. Subsequently, we estimated the G-protein-biased factor of each nalfurafine analog compared to nalfurafine to identify G-protein-biased analogs. The present study calculated G-protein-biased factors using data from the GloSensor^®^ cAMP assay (G-protein-mediated signaling) and PathHunter^®^ recruitment assay (β-arrestin-mediated signaling). As shown in Table 2, the G-protein-biased ratio of SYK-309 was significantly higher than that of nalfurafine (mean ± SEM: 4.46 ± 1.87, *p* = 0.0055, Table 2). These results indicated that, compared to nalfurafine, SYK-309 was the only G-protein-biased κOR agonist among the six nalfurafine analogs.

## 3. Discussion

Here, we evaluated the effects of nalfurafine and six nalfurafine analogs (SYK-160, -186, -245, -308, -309, and -406) on κOR-activated intracellular signaling using the CellKey^TM^, GloSensor^®^ cAMP, and PathHunter^®^ β-arrestin recruitment assays. Our results revealed that all tested compounds activated κOR-mediated intracellular signaling in a concentration-dependent manner as full κOR agonists. Furthermore, most of the EC_50_ values of these test compounds were higher than nalfurafine. In addition, similar results were obtained in our CellKey^TM^ assay and GloSensor^®^ cAMP assay and their correlation seemed to be high. These results suggest that in an impedance assay using the CellKey^TM^ system, the results of impedance changes were reflected mostly with changes in cAMP levels, but not changes in β-arrestin activity. Our previous studies also showed a similar pattern to the present results [61,62]. Our previous study examined the binding affinity of six nalfurafine analogs (the same ones used in the present study) for κORs; nitrogen, with an *N*-cyclopropylmethyl substituent, and 6-amide side chains were indispensable for nalfurafine to bind to κORs, and the phenol ring (3-hydroxy group) was also important for increasing the κOR binding affinity. Compared to our present study, the binding ability *K*_i_ (nM) value of each nalfurafine analog for κORs tended to correlate with the EC_50_ values of CellKey^TM^, cAMP, and β-arrestin recruitment assays [55]. Moreover, the results of one analog, SYK-309, indicated that there was a significant difference in the ratio of G-protein-mediated signaling to β-arrestin-mediated signaling in our present study.

The κOR agonists have been proposed as antinociceptive drugs in humans [24,46,63,64]. These agonists can independently activate multiple signaling mechanisms, making it difficult to screen them using one assay [65]. For this reason, we used three assays in our present study. Schattauer S.S. et al. showed that nalfurafine was a G-protein-biased κOR agonist compared to other κOR agonists such as (-)-U50488H [41]. Nalfurafine exerted anti-scratch and analgesic effects without adverse events such as sedation, motor incoordination, or conditioned place aversion in mice [66]. Many investigations indicated that antinociception induced by κOR agonists is caused by the G-protein-mediated pathway [67], whereas unfavorable effects are caused by the β-arrestin-mediated pathway [68,69,70], suggesting that G-protein-biased κOR agonists could lead to the development of safer and more effective opioid analgesics. Indeed, nalfurafine was initially developed as an analgesic; however, the analgesic and sedative effects were not well separated at analgesic doses [47,54]. Taking the previous studies into consideration, we focused on nalfurafine and investigated the relationships between κOR signaling selectivity and the structural features of nalfurafine using six nalfurafine-based analogs.

Among the nalfurafine analogs in Group A (maintained benzene ring), none had E_max_ values that significantly exceeded those of nalfurafine in both G-protein- and β-arrestin-mediated signaling. In contrast, for G-protein-mediated signaling, SYK-160 (removed the 4,5-ether bridge), SYK-186 (removed the 3-hydroxy group), and SYK-406 (removed both the 4,5-ether bridge and 3-hydroxy group) caused significant increases in EC_50_ values. However, for β-arrestin-mediated signaling, the EC_50_ value of SYK-160 was not significantly changed, whereas those of SYK-186 and SYK-406 were significantly increased. Compared to nalfurafine, the G-protein-biased ratio of SYK-160 was decreased. Still, there was no significant change in the ratios of the Group A compounds (Table 2). Therefore, these data suggest that the 4,5-ether bridge and 3-hydroxy group on the benzene ring were important to activate both G-protein- and β-arrestin-mediated signaling.

Among Group B (converted benzene ring to cyclohexene ring), SYK-245 (removed both the 4,5-ether bridge and 3-hydroxy group) remarkably lost potency in both G-protein- and β-arrestin-mediated signaling compared to nalfurafine; the EC_50_ value increased significantly by ~100 times in the GloSensor^®^ cAMP assay, whereas that in the PathHunter^®^ recruitment assay increased significantly by ~1000 times (Table 1). Furthermore, the G-protein-biased ratio of SYK-245 tended to decrease; however, the change was not significant compared to nalfurafine (Table 2). It is notable that both SYK-308 and -309 significantly increased G-protein- and β-arrestin-mediated signaling compared to SYK-245, as seen in the log EC_50_ value in GloSensor^®^ cAMP assay: SYK-245 vs. SYK-308 (−7.42 ± 0.11 vs. −9.20 ± 0.10, *p* = 0.0001), SYK-245 vs. SYK-309 (−7.42 ± 0.11 vs. −9.13 ± 0.07, *p* = 0.0001); and in the PathHunter^®^ recruitment assay: SYK-245 vs. SYK-308 (−6.91 ± 0.09 vs. −8.20 ± 0.14, *p* = 0.0001), SYK-245 vs. SYK-309 (−6.91 ± 0.09 vs. −7.81 ± 0.08, *p* = 0.0001) (Table 1). These data suggest that the 3-hydroxy group on the cyclohexene ring is important for κOR activities.

Moreover, our present study showed that SYK-309, but not SYK-308, significantly increased the G-protein-biased ratio compared to nalfurafine (Table 2). As a cyclohexene ring is not an aromatic ring, the carbon bound to the 3-hydroxy group is a stereogenic center. Therefore, SYK-308 and SYK-309 possess a 3*R* and 3*S* configuration, respectively. Between SYK-308 and -309, the direction of the 3-hydroxy group was the only structural difference; however, there were significant differences in the EC_50_ values of β-arrestin-mediated signaling (PathHunter^®^ recruitment assay, *p* = 0.031), but not in G-protein-mediated signaling (GloSensor^®^ cAMP assay, *p* = 0.576). Therefore, our present study, for the first time, suggests that the difference in the direction of a hydroxy group at the 3-position of the cyclohexene ring may cause a change in the selectivity of β-arrestin-mediated signaling, and the direction of the 3*S*-hydroxy group could be one of the important key factors for G-protein-biased κOR signaling.

Molecular docking studies of nalfurafine and SYK-186 (removal of the 3-hydroxy group) into κORs have shown that the 3-hydroxy group of the phenolic moiety of nalfurafine interacted with residues Y3.33, K5.39, and H6.52 in κORs via water-mediated hydrogen bonds. In contrast, SYK-186 did not interact with these residues in κORs [57]. However, it is unknown which of the residues in κORs interacts with SYK-309. Therefore, considering the structural features of nalfurafine and the expectation for the development of novel κOR agonists, further investigation with a 3-D docking model using SYK-309 could elucidate the direction of the hydroxy group at the 3-position of nalfurafine analogs that induces G-protein-biased signaling.

There are some limitations in the present study. Since this was an in vitro study, we cannot clinically evaluate the analgesic effects or side effects of the nalfurafine analogs compared to nalfurafine. As a result, we cannot conclude whether SYK-309 can maintain analgesic effects and reduce any unfavorable side effects mediated by the β-arrestin-mediated pathway. In addition, whether G-protein-biased κOR agonists have safer and more beneficial profiles than opioid analgesics is still under discussion. Therefore, further studies in different models are necessary to develop more efficacious opioids without any negative impacts on disorders such as chronic pain and pruritus.

Several studies, particularly by Laura Bohn, claim that arrestin is primarily responsible for the adverse effects of ORs; however, numerous recent studies contradict this assertion [34,71]. Therefore, it is necessary to analyze not only analgesic effects, but also sedative effects of SYK-309, as well as its property as a G-protein-biased agonist against κORs. Moreover, we measured the κOR activities (the G-protein-mediated and β-arrestin-mediated signaling) for only the κOR agonists, the nalfurafine and nalfurafine analogs, in this study. Therefore, it is also necessary that the κOR activities in other morphinan or benzomorphan derivatives which have the A ring modified like the SYK-309 are measured and have their profiles compared with nalfurafine and nalfurafine analogs in the future.

Opioid analgesics, especially μOR agonists, are used to treat pain; however, their usage is sometimes complicated by detrimental side effects [7]. Therefore, developing novel opioids with fewer adverse events is strongly desirable. Recent research indicated that functionally selective κOR agonists elicited neither addictive nor adverse effects [72], and, subsequently, several groups have screened for G-protein-biased κOR agonists [73,74,75,76]. This study showed that the direction of the 3-hydroxy group in nalfurafine is crucial in inducing G-protein-biased signaling. However, to clinically introduce κOR agonists as painkillers, further investigation of the structural properties of nalfurafine—which selectively activate G-protein-mediated pathways via κORs—is necessary.

## 4. Materials and Methods

### 4.1. Chemicals

We used the following regents: nalfurafine—(2E)-N-[(5R,6R)-17-(cyclopropylmethyl)-4,5-epoxy-3,14-dihydromorphinan-6-yl]-3-(furan-3-yl)-N-methlprop-2-enamide), and six nalfurafine analogs that were divided into two groups according to structural characteristics (Figure 1). Group A: SYK-160 (removed the 4,5-ether bridge from nalfurafine), SYK-186 (removed the 3-hydoroxy group from nalfurafine), and SYK-406 (removed the 4,5-ether bridge and 3-hydroxy group from nalfurafine). Group B: SYK-245 (removed the 4,5-ether bridge and 3-hydroxy group from nalfurafine, and converted the benzene ring to a cyclohexene ring), SYK-308 (removed the 4,5-ether bridge from nalfurafine and converted the benzene ring to a cyclohexene ring, with a 3R-hydroxy group), and SYK-309 (removed the 4,5-ether bridge from nalfurafine and converted the benzene ring to a cyclohexene ring, with a 3S-hydroxy group). All chemicals were diluted with dimethyl sulfoxide. All compounds employed were synthesized as described previously [77,78].

### 4.2. Cell Lines

We amplified Halotag^®^-fused κORs (Halotag^®^-κOR, from Kazusa DNA Research Institute, Chiba, Japan) with the pGlosensor^TM^-22F plasmid (pGS22F) from Promega (Madison, WI, USA), following the manufacturer’s instructions. Human embryonic kidney 293 (HEK293) cells were obtained from the American Type Culture Collection (ATCC^®^, Manassas, VA, USA), and stably expressing Halotag^®^-κORs were generated by transfection of the constructed plasmids using the Lipofectamine reagent (Life Technologies, Carlsbad, CA, USA), which were selected based on OR activity measured by the CellKey^TM^ assay or the cAMP assay with Glosensor^®^.

### 4.3. Cell Culture

We cultured HEK293 cells that stably expressed Halotag^®^-κOR/pGS22F in Dulbecco’s modified Eagle’s medium (DMEM) supplemented with 10% fetal bovine serum albumin, penicillin (100 U/mL), streptomycin (100 µg/mL), 700 μg/mL genistein (Glico, Palo Alto, CA, USA), and 100 μg/mL hygromycin in a humidified atmosphere containing 95% air and 5% CO_2_ at 37 °C.

### 4.4. Functional Analysis of ORs Using the CellKey^TM^ System

We examined the effects of nalfurafine and nalfurafine analogs on κORs by the CellKey^TM^ assay system, as described previously [56,79]. We seeded cells at a density of 5.0 × 10^4^ in CellKey^TM^ poly-D-Lysine (Sigma-Aldrich)-coated 96-well microplates with an embedded electrode at the bottom of each well, for 24 h of incubation. After washing with the CellKey^TM^ buffer composed of Hanks’ balanced salt solution (1.3 mM CaCl_2_∙2H_2_O, 0.81 mM MgSO_4_, 5.4 mM KCl, 0.44 mM KH_2_PO_4_, 4.2 mM NaHCO_3_, 136.9 mM NaCl, 0.34 mM Na_2_HPO_4_, and 5.6 mM d-glucose) containing 20 mM 4-(2-hydroxyethyl)-1-piperazineethanesulfonic acid (HEPES) and 0.1% bovine serum albumin (BSA), we incubated the cells for 30 min at 28 °C and treated them with the vehicle or one of the reagents. The change in impedance of an induced extracellular current (dZiec) in each well was measured for 25 min, following a 5 min baseline measurement. The magnitude of change in the dZiec value was defined as ΔZiec. The value for nalfurafine analogs was calculated as a percentage using the highest value for nalfurafine (positive control).

### 4.5. Intracellular cAMP Levels Measured with the GloSensor^®^ cAMP Assay

We performed the GloSensor^®^ cAMP assay as described previously [30,79,80]. In brief, cAMP accumulation was analyzed using cells stably expressing Halotag^®^-KOR/pGS22F. We seeded the cells at 4.0 × 10^4^ cells/well in 96-well clear-bottom white plates (Corning, Corning, NY, USA) and then incubated them for 24 h. After washing the cells with the CellKey^TM^ buffer without BSA, the cells were equilibrated with the diluted GloSensor^®^ reagent at room temperature for 2 h, and the baseline fluorescence intensity was measured for 15 min. After the baseline measurement, cells were treated with the test compounds for 10 min, after which forskolin (3.0 × 10^−6^ M) was added. The fluorescence intensity was measured every 2.5 min for 30 min using Synergy^TM^ H1 (Bio Tek Instruments Inc., Winooski, VT, USA); time-fluorescence curves and the area under the curve (AUC) values of time-fluorescence intensities were calculated. The responses of test compounds were expressed as the AUC of each test compound subtracted from that of the negative control sample (forskolin alone). Data were transformed from each well as the percentage (%) of intracellular cAMP inhibition and calculated by dividing the corrected AUC by those of the standard sample. The standard sample was nalfurafine (10^−7^ M) for Halotag^®^-KOR/pGS22F.

### 4.6. β-Arrestin Recruitment Assay with PathHunter^®^

This was performed as described previously [81]. In brief, U2OS OPRM1, CHO-K1 OPRD1, or U2OS OPRK1 cells were seeded at a density of 1.0 × 10^4^ cells/well in 96-well clear-bottom white plates and incubated for 48 h. The cells were stimulated for 180 min at 37 °C under 5% CO_2_, and the PathHunter^®^ working detection solution was added. The luminescence intensity was measured using the FlexStation 3 (Bio Tek Instruments Inc., Winooski, VT, USA) for 1 h at 25 ± 3 °C. Data are expressed as the maximum signal intensity of each test compound as a percentage of the maximum signal intensity of the positive control.

### 4.7. The Estimated Intrinsic Reactive Activity (RA_i_) and Biased Factors

According to the method developed and refined by Ehlert and colleagues [58,59,60], the G-protein-biased factor was estimated. In brief, each agonist’s intrinsic reactive activity (RA_i_) was estimated by global nonlinear regression analysis [59,82,83]. The RA_i_ of each nalfurafine analog was estimated from the concentration-response curves used to estimate EC_50_ and E_max_ values (Table 1). The G-protein-biased factor was defined as the ratio of the RA_i__ g _value divided by RA_i-B_ (Table 2).

### 4.8. Statistical Analysis and Approval for the Study

Data analyses and concentration-response curve fitting were performed using GraphPad Prism 9 (GraphPad Software, San Diego, CA, USA). Data are presented as means with the standard error of the mean (SEM) for at least three independent experiments. Statistical analysis was performed using a one-way ANOVA, followed by the Bonferroni multiple comparison tests or *t*-tests. A value of *p* < 0.05 was considered statistically significant. All analyses and experiments were approved and performed in accordance with the Guide for Genetic Modification Safety Committee, National Cancer Center, Japan.

## 5. Conclusions

The present study revealed that the direction of the 3-hydroxy group of nalfurafine may be the partial structure inducing G-protein-biased signaling via the weakening of β-arrestin-mediated signaling. Therefore, nalfurafine analogs having a 3“S”-hydroxy group, such as SYK-309, could be considered κOR agonists with a pharmacological profile that selectively activates the G-protein-mediated pathway. This evaluation of the structure–activity relationship is expected to help the development of novel selective κOR agonists.

## Figures and Tables

**Figure 1 molecules-27-07065-f001:**
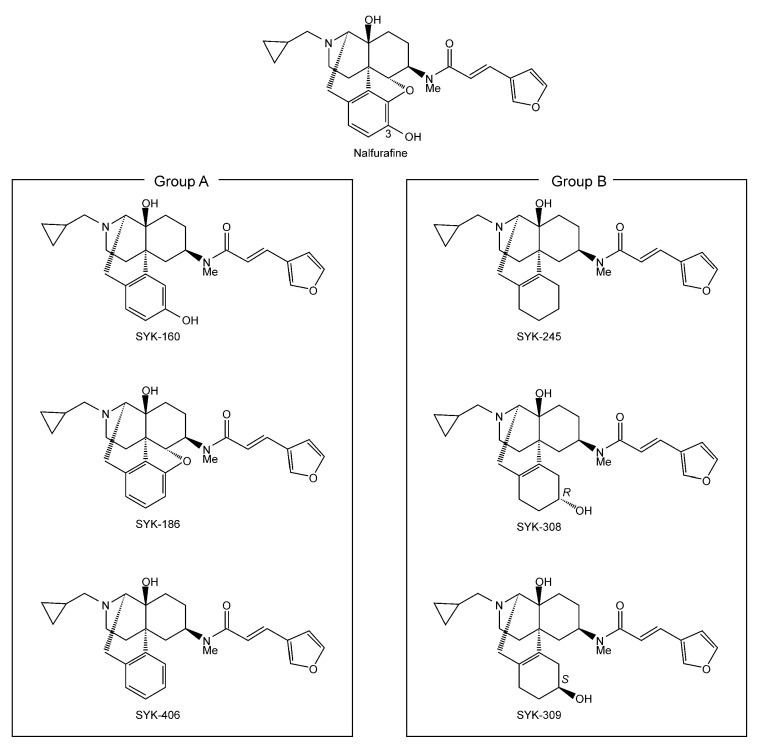
Molecular structures of nalfurafine and six nalfurafine analogs. Nalfurafine analogs were divided into two groups according to their structural characteristics. Group A (SYK-160, -186, and -406) includes nalfurafine analogs with a maintained benzene ring. Group B (SYK-245, -308, and -309) includes nalfurafine analogs with a cyclohexene ring converted from the benzene ring.

**Figure 2 molecules-27-07065-f002:**
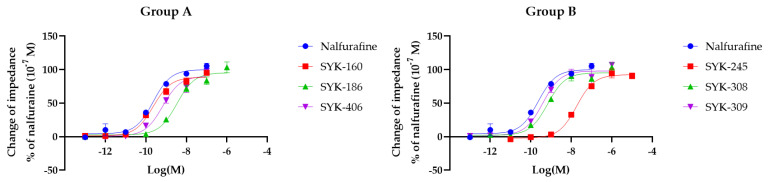
Effect of nalfurafine and six nalfurafine analogs on κORs observed using the CellKey^TM^ system. The cells expressing κORs were treated with nalfurafine (positive control) and six nalfurafine analogs (Group **A**: SYK-160, -186, -406; Group **B**: SYK-245, -308, -309) at concentrations of 10^−13^–10^−5^ M, and changes in impedance (ΔZiec) were measured using the CellKey^TM^ system. Concentration-response curves were prepared by calculating ΔZiec relative to the data obtained for the positive control: 10^−7^ M nalfurafine. All data points are presented as means ± standard error of the mean (SEM) (*n* = 3–6).

**Figure 3 molecules-27-07065-f003:**
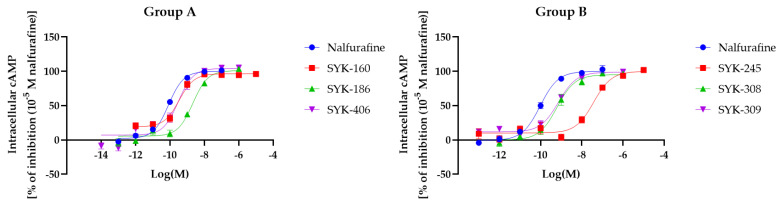
Changes in intracellular cAMP levels induced by nalfurafine (positive control) and six nalfurafine analogs (Group **A**: SYK-160, -186, -406; Group **B**: SYK-245, -308, -309). Cells expressing κORs were treated with the listed compounds (10^−14^–10^−5^ M), and intracellular cAMP levels were measured with the GloSensor^®^ cAMP assay. Concentration-response curves were prepared by calculating cAMP levels relative to the data obtained with 10^−7^ M nalfurafine. Data are presented as means ± SEM (*n* = 3–9).

**Figure 4 molecules-27-07065-f004:**
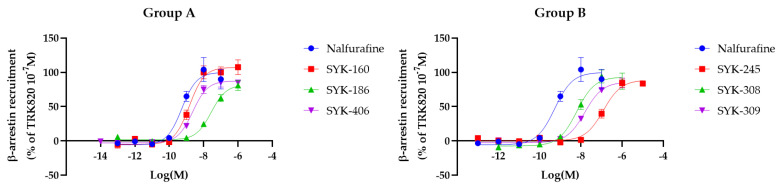
Levels of β-arrestin recruitment through κORs induced by nalfurafine and six nalfurafine analogs. We performed a PathHunter^®^ β-arrestin assay in cells expressing κORs by treatment with nalfurafine (positive control) and the six nalfurafine analogs (Group **A**: SYK-160, -186, -406; Group **B**: SYK-245, -308, -309) at concentrations of 10^−14^–10^−5^ M. Concentration-response curves were prepared by calculating intracellular β-arrestin levels relative to the data obtained for nalfurafine (positive control: 10^−7^ M). All data points are presented as means ± SEM (*n* = 5–8).

**Table 1 molecules-27-07065-t001:** E_max_ and log EC_50_ values for nalfurafine and six nalfurafine analogs in the CellKey, GloSensor cAMP, and PathHunter assays for κORs.

		*CellKey Assay*		*GloSensor cAMP Assay*		*PathHunter Assay*	
	Compounds	Log EC_50_ (M)	E_max_ (%)	Log EC_50_ (M)	E_max_ (%)	Log EC_50_ (M)	E_max_ (%)
	Nalfurafine	−9.64 ± 0.10	100.00 ± 3.09	−10.09 ± 0.06	100.00 ± 1.74	−9.27 ± 0.19	100.00 ± 7.43
*Group A*	SYK-160	−9.64 ± 0.90	88.74 ± 2.55	−9.47 ± 0.10 ***	96.51 ± 1.97	−8.83 ± 0.12	107.27 ± 4.64
	SYK-186	−8.47 ± 0.10 ***	95.42 ± 3.18	−8.67 ± 0.09 ***	101.14 ± 2.74	−7.56 ± 0.11 ***	81.76 ± 3.83
	SYK-406	−9.17 ± 0.12 **	90.33 ± 3.94	−9.52 ± 0.12 ***	104.26 ± 3.44	−8.64 ± 0.09 **	87.47 ± 2.77
*Group B*	SYK-245	−7.74 ± 0.07 ***	92.51 ± 2.36	−7.42 ± 0.11 ***	103.21 ± 3.60	−6.91 ± 0.09 ***	88.78 ± 3.35
	SYK-308	−9.16 ± 0.09 **	95.41 ± 2.57	−9.20 ±0.10 ***	98.16 ± 3.21	−8.20 ± 0.14 ***	93.29 ± 5.51
	SYK-309	−9.41 ± 0.12	98.06 ± 3.54	−9.13 ± 0.07 ***	102.03 ± 2.16	−7.81 ± 0.08 ***	85.92 ± 3.08

Nalfurafine was used as the positive control. E_max_ (%) and log EC_50_ (M) values (means ± SEM) were calculated according to the results shown in Figure 2, Figure 3 and Figure 4. Statistical comparisons were made using GraphPad Prism 9 software and are expressed as means ± SEM. Differences between the means were analyzed with one-way analysis of variance (ANOVA) or *t*-tests. One-way ANOVA was followed by Bonferroni post hoc analysis. Significant levels are ** *p* < 0.01 and *** *p* < 0.001 compared with nalfurafine. The number of samples of EC_50_ and E_max_ are indicated as follows: *n* = 3–6 (the CellKey^TM^ assay), *n* = 3–9 (the GloSensor^®^ cAMP assay), and *n* = 5–8 (the PathHunter^®^ recruitment assay).

**Table 2 molecules-27-07065-t002:** G-protein-biased factors of nalfurafine analogs for G-protein and β-arrestin coupling.

Compounds	G-Protein-Biased Ratio (Mean ± SEM)	*p*-Value
Nalfurafine	1	n.s.
SYK-160	0.36 ± 0.05	n.s.
SYK-186	1.47 ± 0.30	n.s.
SYK-406	0.89 ± 0.18	n.s.
SYK-245	0.35 ± 0.67	n.s.
SYK-308	1.83 ± 0.80	n.s.
SYK-309	4.46 ± 1.87 **	0.0055

The parameters were calculated from the same agonist concentration-response curves used to estimate EC_50_ and E_max_ values in Figure 2 and Figure 3, and in Table 1, using the method described by Ehlert and colleagues [58,59,60]. The prototype of the selective κOR agonist, nalfurafine, was designated as a standard reference ligand. The bias factor of G-protein signaling for a given ligand is defined as the ratio of the intrinsic activity (RA*_i-G_*) divided by RA_i-b_. The G-protein-biased ratios (means ± SEM) were calculated according to the results shown in Table 1, ** *p* < 0.01, compared to bias factor of 1 by *t*-test.

## Data Availability

Not applicable.

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
