# Peer review of "Evaluation of the Intracellular Signaling Activities of κ-Opioid Receptor Agonists, Nalfurafine Analogs; Focusing on the Selectivity of G-Protein- and β-Arrestin-Mediated Pathways"

_molecules, 2022, doi:10.3390/molecules27207065_

Round 1

Reviewer 1 Report

Dear Authors:

The authors attempted to separate the G-protein and b-arrestin activities of the KOR agonist nalfurafine analogs and succeeded in doing so in this very interesting article. I therefore recommend publication of this manuscript in Molecules.
However, the following points need to be discussed regarding the mechanism.

Are there any examples of replacing the benzene ring of the A ring with a cyclohexene ring and measuring KOR activity in morphine-related morphinans or benzomorphans? If so, the results should be compared and discussed.

In the X-ray structure of the antagonist JDTic in complex with KOR (PDB ID: 4DJH), the 3-position hydroxyl group of the benzene ring is hydrogen bonded to the side chain hydroxyl group of Tyr139 and the main chain carbonyl oxygen atom of Lys227 via two water molecules (see link below). If this benzene ring is replaced by a cyclohexene ring, is the 3S-hydroxyl group more convenient to maintain the hydrogen bonding network?

PDB ID: 4DJH
https://www.rcsb.org/3d-view/4DJH?preset=ligandInteraction&label_asym_id=C

Minor corrections:
Periods are missing in the last sentences of Figures 2 and 3.

Reviewer 2 Report

In this manuscript, the authors continue their investigation of structure-activity relationships of nalfurafine analogues as biased kappa opioid receptor ligands. The manuscript focuses on the pharmacological characterization of these compounds as they have described the synthesis previously. A certain problem is that the authors are apparently unaware of two recent developments in the GPCR field: (1) there seem to be no signaling pathways that are transduced solely via arrestin without any G-protein involvement (see for example Grundmann et al., Nat. Commun. 9: 341 (2018) but also papers by the Gutkind group) (2) the idea that, for opioid receptors, G-protein signalling is “good” and arrestin signalling is “bad”, is probably wrong except for receptor desensitization (this has been shown mostly for the mu opioid receptor, e.g. by Kliewer et al., Br. J. Pharmacol. 177: 2923-2931 (2020)). I have also a few suggestions on how to improve the language of the manuscript.

line 48: what is “activating opioid receptor”?

line 59-60: my suggestion would be “about 20% of patients with chronic pain using opioids were abusing them”

line 77-78: although there is a number of old papers, particularly by the group of Laura Bohn, claiming that arrestin is solely responsible for the side effects of mu opioid receptors, there is newer evidence that this is not true (see above).

line 90: references 28-33 deal with the mu opioid receptor and can therefore not be used to suggest that arrestin is causing dysphoria via the KOR

line 96: “pruritis” should probably read “pruritus”

section 2.1: what is the KOR expression level in these cells?

section 2.2: what is the KOR expression level in these cells?

sections 2.1 and 2.2: the authors could comment on the high correlation between the CellKey and the GloSensor signals. This correlation seems to be very high which suggests that label-free methods such as CellKey do not detect arrestin signal but just G-protein-mediated signals. Other authors have published similar results.

section 2.3: what is the KOR expression level in these cells? This repeated question becomes relevant because receptor reserve plays some role in comparing the effects of ligands on G-protein activation and arrestin binding. G-protein activation and label-free methods have readouts that undergo at least one amplification step whereas there is no such amplification in receptor-arrestin interactions. For example, SYK-245 seems to be a full agonist in the CellKey and cAMP assays but not in the PathHunter assay. Receptor reserve also affects EC50 levels, so it is expected that G-protein mediated signals have lower EC50 values than arrestin-mediated signals if receptor reserve is high (see next point).

line 182: “EC50 values … were increased” – compared to what?

Table 1: please supply n for each set of data (i.e. EC50, Emax, n).

section 2.4 and Table 2: looking at the raw data, I consider it fairly unlikely that SYK-308 is not a biased ligand whereas SYK-309 is. The curves for the cAMP levels are practically superimposable, so the difference must come from the PathHunter assay. I notice that the EC50 are significantly different whereas the Emax are not, but the difference is less than threefold. It is also noticeable that the bias factor of nalfurafine is 1. Is this set arbitrarily, or is it a result of the calculations? Does the method of Ehlert work only by comparing compounds to a reference ligand?

line 281: my suggestion would be “the EC50 value of SYK-160 was not significantly changed”

lines 288-290: my suggestion would be “SYK-245 (…) remarkably lost potency in both G protein- and beta-arrestin-mediated signaling”

References formatting: many references have random italics in their title
